# Characterization of Depressive Symptom Trajectories in Women between Childbirth and Diagnosis

**DOI:** 10.3390/jpm12040538

**Published:** 2022-03-28

**Authors:** Natalia Chechko, Susanne Stickel, Elena Losse, Aliaksandra Shymanskaya, Ute Habel

**Affiliations:** 1Department of Psychiatry, Psychotherapy and Psychosomatics, Faculty of Medicine, RWTH Aachen University, 52074 Aachen, Germany; sstickel@ukaachen.de (S.S.); ellosse@ukaachen.de (E.L.); ashymanskaya@ukaachen.de (A.S.); uhabel@ukaachen.de (U.H.); 2Institute of Neuroscience and Medicine, JARA-Institute Brain Structure Function Relationship (INM-10), Research Center Jülich, 52428 Jülich, Germany; 3Institute of Neuroscience and Medicine, Brain & Behavior (INM-7), Research Center Jülich, 52428 Jülich, Germany

**Keywords:** postpartum depression, adjustment disorder, temporal development, depressed mood: maternal attachment

## Abstract

The inhomogeneity of postpartum mood and mother–child attachment was estimated from immediately after childbirth to 12 weeks postpartum in a cohort of 598 young mothers. At 3-week intervals, depressed mood and mother–child attachment were assessed using the EPDS and the MPAS, respectively. The diagnosis was based on clinical interviews at the end of the 12-week follow-up. The latent class mixed model estimated multiple distinct patterns in depressed mood and mother–child attachment. The baseline EPDS cluster contained 72% of the study population and showed low EPDS values during the follow-up period, while the five remaining clusters showed either deterioration or improvement of the EPDS levels. The majority of women with postpartum depression showed deteriorating, and the majority of adjustment disorder cases improving, behavior. While the cases with more pronounced EPDS values were found to constitute more homogeneous clusters in terms of diagnosis, subclinical or only temporarily increased EPDS levels represented less homogeneous clusters. Higher EPDS levels correlated with the higher risk factor profiles. The four MPAS/EPDS clusters demonstrated that higher EPDS lead to lower mother–child attachment, and vice versa.

## 1. Introduction

The time around childbirth is frequently associated with changes in mood levels in young mothers. These changes, however, are not homogeneous, and represent a range of perinatal mood conditions. About 30–80% of new mothers experience postpartum blues [1], which, in most cases, is normal adaptation to hormonal withdrawal in the early postpartum period (particularly, sudden changes in the estradiol levels) [2].

While the “blues” usually resolve spontaneously, typically within the first few days postpartum, symptoms of sadness lasting longer than two weeks ought to be regarded as postpartum depression [2], or, in some cases, as postpartum adjustment disorder (AD) [3].

Postpartum depression (PPD) occurs in 9–15% of childbirths and represents a depressive disorder with peripartum onset (around delivery or within the first four weeks postpartum, [4]). As opposed to PPD, AD is recognized as a stress-response syndrome or a maladaptive reaction to an identifiable stressor (DSM-V), which can be observed in 10–15% of postpartum women [3]. The crucial difference between postpartum AD and PPD is that the severity of symptoms of the former does not meet the criteria for depression at any time point [4]. Therefore, AD represents a subthreshold form of depressive or anxious mood. A clear differentiation between AD and PPD, particularly in the first postpartum weeks, is not always possible [3]. While less common than baby blues or postpartum AD [3,5], PPD has the potential to significantly affect the health of both mother and child [6]. It influences a wide range of outcomes for offspring, including infant growth, physical health, nutritional status, executive functions, socioemotional development, academic achievement, as well as the risk for psychopathology [7,8,9,10,11,12,13,14]. In particular, there is an increased risk for a transmission of intergenerational depression [14,15,16]. Therefore, early biomarkers, predictors, and early interventions are important for the wellbeing of both mothers and their children. However, the reality is that PPD is often overlooked during the postnatal visits, resulting in lost time and opportunities as preventative interventions are most effective if administered shortly after delivery [17,18,19].

Given that the changes in perinatal mood have different origins, the heterogeneity of the associated trajectories of depressive mood symptoms is not surprising. Campbell and colleagues followed 1261 mothers when children were 1, 6, 15, 24, 36, and 54 months and in the first grade in [20]. Based on a longitudinal assessment of the Center for Epidemiological Studies Depression Scale (CES-D) [21], around 14% of participants showed chronically elevated levels of depression at least in the first year following childbirth. The more serious and/or chronic depression symptoms were consistently associated with less sensitive behavior toward the child. Qandil et al. [22] focused on the heterogeneity of depressive symptoms in 101 new mothers in the first 6 months postpartum, with the assessments being made first 1 week after delivery, and then 2 weeks, 6 weeks, 3 months, and 6 months later. The depressive symptoms scores in the last weeks of pregnancy were assessed retrospectively. The assessments included the Edinburgh Postnatal Depression Scale (EPDS), with PPD defined by depressive symptoms (EPDS score ≥ 11) at ≥2 follow-up time points. Based on that criterion, depression at some point of the follow-up was suggested to affect up to 50% of the sample. In 30% of the participants, the depressive symptoms developed most frequently within 1 month postpartum, thus indicating an early onset of the condition. About two-thirds of the early-onset depressive symptoms were likely to decline by 7 weeks post-delivery, with about one-third of them persisting beyond this point and being still detectable at 6 months postpartum. When the mothers developed depression after the first 4 weeks postpartum (late onset, about 11% of the sample), they were likely to still have symptoms at 6 months after childbirth. Finally, 6% of the sample (the chronic group) reported above-threshold or close-to-threshold scores already before pregnancy and for the entire duration of the follow-up period. In [23], a large sample of 12,121 women was investigated at 18th and 32nd weeks of gestation, and between the 8th week and the 61st month after childbirth, using seven EPDS assessments. Similar to the study by Qandil, in [22], when the depressive symptoms were detected already in pregnancy (11% of the sample), they tended to persist up to 5 years postpartum. In addition, about 8% of participants had elevated levels of depressive symptoms during pregnancy that declined around 8 weeks after delivery. Finally, a group of participants (4%) with no symptoms during pregnancy showed depressive symptoms at 8 weeks postpartum, which persisted till 5–6 years following childbirth. Given that the first assessment was conducted only at 8 weeks postpartum, it was not possible to draw any conclusion regarding the onset time of depression in relation to childbirth.

In sum, these studies vary widely in terms of time points, ranging from weeks to months and years, and from being antenatal to postnatal, lacking focus on a short-term trajectory closely assessing the early postpartum period. The early postpartum period (10 to 19 days after childbirth in particular) is a time of the highest risk for women to develop mental disorders [24], with the most frequent one being PPD. Women with PPD are at an increased risk of developing chronic psychiatric conditions. For instance, 50% of women with a history of PPD have been found to develop at least one further episode of depression [25]. Therefore, it is critically important to monitor women’s health within the first few weeks of childbirth, beginning as close to the point of delivery as possible. Early detection, intervention, and appropriate treatment of PPD can help prevent chronification, reducing the emotional and financial burden of the disease [26]. In addition, in all of these studies, the PPD characterization is based only on self-report assessments, explaining the high prevalence of depression [27], with none of them considering AD as a separate diagnosis, which is recognized as a stress-response syndrome. As a subthreshold diagnosis, it is defined as a maladaptive reaction to an identifiable stressor (DSM-V). Thus, the crucial difference between postpartum AD and PPD is that the severity of the AD symptoms does not meet the criteria for depression at any time point. Therefore, in clinical practice, both baby blues and AD need to be considered as important differential diagnoses of PPD. Neither condition has the tremendously debilitating effect of a clinically manifest depression and, normally, neither requires treatment. A timely differentiation between PPD and AD can aid the identification of the most affected and vulnerable cases. However, subclinical depression or adjustment disorder may, in some cases, be a significant public health issue because of its high prevalence in the population [28]. In some cases, women suffering from these conditions are likely to benefit from relevant supportive treatment. Furthermore, the information about risk factors (e.g., psychiatric history, baby blues or stressful life events (SLE)) is frequently missing, and only one study included maternal attachment in relation to the dynamics of depressive mood [20]. Risk factors provide crucial insight into the possible development of PPD due to their association with the prevalence of the diagnoses. The presence of multiple risk factors is a likely indication of stronger depressive symptoms [3].

The main aim of the present study was to investigate the dynamics and heterogeneity of depressive symptoms in clinically diagnosed PPD and AD cases, determining the differences between the diagnoses in terms of their temporal development and dynamic course. The follow-up was conducted by means of, on the one hand, a fine-mesh approach and, on the other, an independent clinical interview performed after 12 weeks. This two-pronged approach facilitated unbiased diagnoses of PPD and AD. The temporal development of the diagnoses was accomplished by identifying trajectories of maternal depression symptoms (EPDS) and mother–child attachment (Maternal Postnatal Attachment Scale; MPAS) from immediately after childbirth to 12 weeks postpartum using the latent class mixed model (LCMM) approach. The five-time point longitudinal design allowed us to estimate the short-term inhomogeneity of (i) depressive symptoms, (ii) mother–child attachment, and (iii) the interaction between depressive symptoms and maternal attachment. The association between demographic and anamnestic factors and the estimated clusters was evaluated.

Based on the available literature, we assumed that the EPDS trajectories would form several distinct temporal clusters, and that the three postpartum states (PPD, AD, and healthy controls (HC)) would be distributed across these clusters, with some clusters containing trajectories predominately associated with one of the three postpartum states. The diagnoses were not involved as variables in the calculations performed for longitudinal cluster analysis. However, we also expected some inhomogeneity in terms of the diagnosis distribution in these clusters. We expected the trajectories with high EPDS scores to correlate with higher risk profiles and higher numbers of PPD diagnosis. Based on our previous work (i.e., [3]), which showed a correlation between MPAS and EPDS, we expected an inverse correlation between the trajectories of mother–child attachment and depressive mood. The investigation of this latent correlation should deliver common latent clusters, characterized by the interplay between EPDS and MPAS, contributing in turn to the classification of PPD.

## 2. Materials and Methods

### 2.1. Participants

As part of the ongoing longitudinal study (Risk for Postpartum Depression (RiPoD) study) aimed at early recognition of PPD, 643 postpartum women were recruited at the Obstetrics ward of the university hospital within 1–6 days of childbirth between November 2015 and July 2020. Women with depression at the time of recruitment, severe birth- and pregnancy-related complications (e.g., eclampsia, HELLP (Hemolysis, Elevated Liver enzymes and Low Platelets)), alcoholic or psychotropic substance dependency or use during pregnancy, history of psychotic or manic episodes, and antidepressant or antipsychotic medication during pregnancy, were excluded from the study. In addition, only mothers of healthy children (determined by the routine German Child Health tests (U2) conducted within the first 3–10 days of life) were included. Prior to enrolment in the study, written informed consent was obtained from each participant. The study protocol was drafted in accordance with the Declaration of Helsinki and approved by the Institutional Review Board of the Medical Faculty, RWTH Aachen University (approval number: EK-208/15, date of approval: 13 January 2021).

### 2.2. Procedure and Questionnaires

The study involved follow-up interviews over a period of 12 weeks after childbirth, with the data being acquired at five time points. Assessments were made at the study center one to six days postpartum (T0) and 12 weeks postpartum (T4), and via remote online questionnaire at 3 weeks (T1), 6 weeks (T2), and 9 weeks (T3) postpartum.

At T0, the clinical-anamnestic screenings (demographic information, information about the pregnancy as well as individual and family psychiatric history) were carried out and the current depressive symptoms after childbirth were assessed using the Edinburgh Postnatal Depression Scale (EPDS) [29], a 10-item self-report instrument. A cut-off score above 10 indicated symptoms of depression as validated and recommended for a German sample [30]. A record of the number and nature of stressful life events was obtained through the Stressful Life Events Questionnaire (SLESQ) [11], which includes possible encounters with 11 traumatic events. Symptoms of premenstrual syndrome (PMS) were assessed with the Premenstrual Tension Syndrome Scale. Next to the total PMS score, cut-off scores were used to identify those with no symptoms (0–4), moderate symptoms (5–13), and severe symptoms (>14). At T1-T4, participants were required to log into the online survey (“Survey Monkey” software) to help assess the preceding 3 weeks by means of the EPDS and the Maternal Postnatal Attachment Scale (MPAS) [31], a 19-item self-report measure of mother–child attachment and its quality (hostility toward the child or pleasure in interaction with the infant).

In addition, at T1, baby blues symptoms during the first few postpartum weeks were assessed using the Maternity Blues Questionnaire (MBQ; [32]). After 12 weeks of participation (T4), the mothers were invited for a final clinical interview, during which information was sought regarding mood, general wellbeing, subjective quality of support at home and breastfeeding behavior. Based on the DSM-V criteria, participants with depressive mood were assigned either to the PPD group or the AD group. Those who remained non-depressed were assigned to the HC group.

### 2.3. Missing Data

In total, 643 women were interviewed for the study. For our analysis, patients were removed from the cohort based on (1) unclear diagnosis and (2) the proportion of missing values. In the first iteration, we performed data cleaning by removing variables with more than 80% missing values in anamnesis data. Next, we removed patients with missing MPAS or EPDS scores, as they were subsequently used as dependent variables for class assignment, which left 598 participants for further analysis. From 36 acquired parameters (pre- and postnatal features), 25 had missing values, which we considered for imputation. We looked closely at participants with more than 5% of missing values in variables (Appendix A to determine whether to impute the missing values or remove the subjects). Variables with more than 5% of missing values were “Support at home”, “PMS Value”, and “PMS Severity”. We noticed that the data for these variables were missing mainly in the beginning of data acquisition, which might have been due to mistakes in the acquisition process. Therefore, we considered these data to be missing at random. The data on complications during pregnancy and childbirth were also missing in the beginning of the second half, again likely due to interviewer error, and, therefore, considered missing at random. After inspecting the frequency of missing data with MICE package (RStudio, Version 1.3.1093, Boston, MA, USA [33,34,35]), we used multiple imputation from MICE package to complete the missing data. For different kinds of variables, the following imputation methods were used: for numerical data–pmm, for ordered–polr, for binary–logreg, for factor–polyreg.

### 2.4. Parameter Overview

Correlation of imputed features among themselves was calculated with Cramér’s corrected statistics with threshold 0.8 to reduce feature space. To identify diagnosis differences, non-parametric tests were used, i.e., the chi-squared test for categorical variables, the Wilcoxon rank-sum test for continuous variables, and the Kruskal–Wallis test for ordered variables.

### 2.5. Methods: LCMM

The characterization of longitudinal trajectories was performed in compliance with the GRoLTS checklist [35] using a latent class mixed model (LCMM) approach from [36,37]. The software used for the analysis is lcmm and multlcmm package of RStudio, Version 1.3.1093, Boston, MA, USA [33,36]. LCMM allows the estimation of heterogeneous trajectories that are defined by the overall course of a latent process using ML framework (maximum likelihood). In the LCMM approach, latent class models are used to find homogenous latent groups of subjects (population level). A mixed model is used to describe the mean trajectory over time in each latent group (class-specific component). Furthermore, the individual correlation between repeated measures is taken into account (individual-specific component) [38,39].

In order to determine the optimal number of trajectories, we compared several models with an increasing number of trajectory classes. The models were defined as combinations of quadratic and linear terms (11 models pro trajectory). Furthermore, we investigated the stability of models by using different random input conditions (grid search with the number of departures from random initial values being 100 and the number of iterations for the optimization algorithm being 50). Furthermore, for each model, we investigated the effect of a diagonal versus unstructured random-effect covariance matrix and a class-specific versus proportional random-effect covariance matrix (four possibilities). We used several indicators to select the best model: the Akaike information criterion (AIC), the sample-adjusted Bayesian information criterion (SABIC), and the Lo–Mendell–Rubin likelihood test LMR LRT [40]. The lower values corresponded to the better model fit, and the LMR LRT test between the models with different number of classes should be significantly different (*p* < 0.001) Furthermore, we considered entropy (higher the value the better class composition), and the percentage of participants within each class. Since we expected one class to contain patients with extremely pronounced symptoms, we assumed the lowest percentage in the class to be 3%, in contrast to the normally assumed threshold of 5% in [41]. The full tables with model characteristics can be provided upon request. We did not include covariates in the LCMM model (so-called standard three-step model), since, according to the suggestion of GRoLTS, it would lead to possible contamination of class determination [35]. The same approach was used in [42]. In our analysis, EPDS (first approach) served as a time factor for LCMM estimations. EPDS data were investigated upon normality using the D’Agostino, Shapiro–Wilk and Anderson–Darling tests. If one of the tests failed, data were assumed non-uniform. Test results of normality of the EPDS values are reported for increased reproducibility of this paper in Appendix A. The non-normally distributed data were fitted using Fleishman power method [43]. In several cases, coefficients could not be determined, further fitting was performed with 101 continuous distributions supplied by scipy.stats [44], and the best fit was determined based on chi fit goodness. The comparisons between the clusters as to how their risk factor profiles differed were performed using chi-squared and Kruskall–Wallis tests.

### 2.6. Methods: MultLCMM

MPAS was also acquired at 3-week intervals, although beginning 3 weeks after childbirth, until the last interview when the participants were accessed by the clinician and diagnosed. Thus, the data were equidistantly spaced between 3 and 12 weeks (T1–T4). In the combined analysis of MPAS and EPDS (second approach) analysis, we used multlcmm function from the LCMM package from RStudio, Version 1.3.1093, Boston, MA, USA [33]. MPAS and EPDS were considered to be multivariate markers for the same latent processes on time scale from T1 to T4. To achieve comparable scale behaviors, EPDS values were reversed, i.e., rescaled as newEPDS = max (oldEPDS) − oldEPDS + 1, so that high values on the new scale corresponded to the low old values, thus helping improve model convergence.

## 3. Results

### 3.1. Population

The cohort consisted of 598 women, of whom 90 were diagnosed with AD (15%) and 55 with PPD (9%), and the rest of the participants remained non-depressed (HC: 76%). Significant differences (*p* < 0.001) were observed in several categories between features and the diagnosis. Compared to their healthy counterparts, participants diagnosed with PPD or AD had lower income, either a family or personal psychiatric history, higher premenstrual syndrome (PMS) score, more SLE, and more experience of depression both related and unrelated to previous pregnancies. More often than not, women with PPD and AD experienced baby blues and birth-related traumas following childbirth. The *p*-values are provided in Appendix A.

### 3.2. LCMM EPDS: Model Selection

Latent class mixed model (LCMM) was used to estimate the temporal trajectories of EPDS scores acquired in the 3-week period between childbirth and the final interview when the participants were diagnosed. Thus, the data were equidistantly spaced between 0 and 12 weeks (T0–T4). The EPDS scores were characterized as low (<10), subclinical (10 ≤ EPDS < 12) and clinical (≥12). The model selection procedure is described in Appendix A (LCMM EPDS: Model Selection), while the technical details of estimated models are provided in Appendix A. All models were found to converge, remaining stable under disparate starting conditions. However, for 6 clusters, only a few models converged under the variable starting conditions, or possessed not zero classes, hence model stability was considered important in model selection in 6 cluster cases. To determine the number of clusters and the relevant descriptive model, we considered the form and overlap of trajectories (Appendix A), selecting the six-cluster model as the final model because it better represented the critical cases with high or growing EPDS. Furthermore, the increase in entropy was significant.

### 3.3. LCMM EPDS: Clusters

Based on the EPDS scores, the LCMM model delivered six clusters (Figure 1), which are further discussed with respect to the heterogeneity of diagnoses in each cluster (in percentage) as well as in terms of prenatal feature differences. The largest cluster (cluster 5, *n* = 428, 72%) was found to have stably low EPDS at all time points and consisted of 97% of HC and 3% of AD/PPD, and was, therefore, considered as baseline. The majority of the HCs in the sample (92%, *n* = 415) appeared in the baseline cluster (Figure 1f).

### 3.4. Recovering Clusters

Three clusters (clusters 1, 3, and 6 of the LCMM model) were characterized by the reduction in EPDS value with time (Figure 1a,c). Showing subclinical EPDS values in the beginning of the study, LCMM cluster 1 (67% AD, 30% HC, and 3% PPD) recovered with time, although remained significantly higher than those of the baseline cluster throughout the observation period. LCMM cluster 3 (68% AD, 29% HC, and 3% PPD) initially demonstrated clinically elevated EPDS values, which recovered with time and finally reached the baseline levels. LCMM cluster 6 (65% AD, 20% HC, and 15% PPD) was characterized by a delayed onset of the depressive symptoms with EPDS values being at baseline levels in the beginning of the study and rising to subclinical/clinical levels between weeks 3 and 6, and thereafter receding to the baseline cluster level at the end of the follow-up (Figure 1c). Like the AD cases in the recovering clusters, the PPD cases also showed improvement over time. Barring the EPDS values at 3 weeks postpartum (for HC: 7.6 (standard deviation SD = 1.8), for PPD: 13.0 (SD = 5.7), and for AD: 11.2 (SD = 3.3)), which were higher for PPD and AD compared to HC, there were no differences between the different diagnostic categories in the recovering clusters.

There were no differences among the initially clinical/subclinical recovering clusters with respect to the risk factors; however, in comparison to the stable low baseline cluster, these clusters had increased risk profiles (please refer to Table 1). In addition, cluster 1 (increased subclinical EPDS levels in the first three postpartum weeks) and cluster 6 (delayed onset, clinically elevated EPDS levels between weeks 3 and 6) had higher numbers of birth-related psychological and physical traumas compared to the baseline cluster. Furthermore, women in the initially clinical/subclinical recovering clusters reported more stressful life events (SLE) compared to their counterparts in the delayed onset cluster (*p* = 0.003). All information including *p*-values and cluster-wise comparison is provided in Appendix A.

### 3.5. Deteriorating Clusters

Two clusters (clusters 2 and 4 of the LCMM model) were characterized by an increase in EPDS values during the 12-week period (Figure 1b). While LCMM cluster 2 (100% PPD) had clinically elevated EPDS levels from the beginning of the follow-up, LCMM cluster 4 (64% PPD, 20% AD, 16% HC) demonstrated initially subclinical or baseline EPDS levels with a gradual increase to subclinical/clinical levels around the 3rd week. The subjects in cluster 2, characterized by clinically elevated EPDS values from the beginning of the follow-up, had significantly more psychiatric diagnoses in previous pregnancies and had lower income compared to their counterparts in cluster 4 (Appendix A). Almost half of the PPD cases assigned to cluster 2 had psychiatric diagnoses in previous pregnancies compared to 3% in cluster 4 (Appendix A). Both clusters showed an increased risk profile in comparison to the baseline EPDS cluster, e.g., significantly higher PMS symptoms, more experiences of previous depression, stressful life events (SLE), baby blues, and lower support at home (Table 1 and Appendix A).

### 3.6. Trajectories of AD Cases within the Recovering Clusters

Seventy-nine percent of all AD cases belonged to one of the recovering clusters (Figure 1e), with the subjects showing a recovery pattern (improvement in EPDS) by the end of the 12-week follow-up. The majority of the participants with AD (65% of all cases) had subclinical or clinically increased EPDS values only in the first 3 weeks of the follow-up, most of them belonging to cluster 1. The remaining 14% of women affected by AD (cluster 6) showed a small increase in EPDS levels after around 3 weeks and normalization after the 6th week.

### 3.7. Trajectories of PPD Cases within the Deteriorating Clusters

The two deteriorating clusters (Figure 1d) included 84% of all PPD cases with the subjects differing in income level, psychiatric diagnosis in previous pregnancy, as well as EPDS values at all time points. Thus, the subjects with later PPD and clinically elevated EPDS values at T0 (31% of all PPD cases in the sample) were found to be more severely affected by depressive symptoms at all time points of the follow-up.

### 3.8. Results MultLCMM MPAS + EPDS: Model Selection

A significant EPDS value difference between the clusters was reflected in significant differences in MPAS values (Table 1). Therefore, we sought to investigate the latent classes common to the MPAS and EPDS scores. To that end, we used the MPAS scores at four time points (3, 6, 9, and 12 weeks postpartum) and combined them with the EPDS scores at the same time points. The best multlcmm [20], which discovered the latent classes common to the MPAS and EPDS scores with the lowest values of SABIC and AIC and high entropy, are provided in Appendix A. Cluster models 5 and 6 were both unstable under varying starting conditions. The four-cluster model was selected based on LMR LRT tests. The appropriate MPAS and EPDS trajectories for the selected four-cluster model are shown in Figure 2. The selected model was stable and possessed the lowest criteria values. For MPAS, no clinically significant threshold has been found [29]. However, in [30], a mean score of 84.6 at 4 months postpartum was reported. Therefore, to simplify the reference, we defined MPAS ≤ 85 as low and MPAS > 85 as high MPAS, the latter suggesting more sensitive maternal behavior toward the child.

### 3.9. MultLCMM MPAS/EPDS: Clusters

The most homogeneous cluster (*n* = 185, Figure 2a,c; cluster 2), which consisted 99% of HCs and had stably low EPDS and high MPAS at all time points, was taken as the baseline. Compared to the other three MPAS/EPDS clusters, this cluster had significantly lower risk factors (Appendix A). Kruskal–Wallis and chi-squared tests were conducted to estimate significant differences in covariates between the baseline and the three MPAS/EPDS clusters (Appendix A). MPAS/EPDS cluster 1, which was the largest (*n* = 278), consisting of 89% HC and 11% AD (Figure 2a,c), once again demonstrated stable, low EPDS and high MPAS scores.

The remaining two MPAS/EPDS clusters were termed upright U-shaped (*n* = 68, Figure 2b,d; cluster 3) and inverted U-shaped (*n* = 67, Figure 2b,d; cluster 4), the former showing a reduction in clinical EPDS scores to subclinical levels by the 9th week, after which it had started to slightly increase. However, except for the 3rd week after childbirth, the EPDS values remained low. The MPAS scores, on the other hand, increased at week 6 and remained constant, though lower than 85. This cluster mainly comprised AD (57% AD, 12% HC, 31% PPD).

The inverted U-shaped cluster consisted of mainly PPD (51% PPD, 28% AD, 21% HC) subjects, in whom EPDS values rose to subclinical/clinical levels at week 6 and remained elevated until the 9th week. Although the EPDS values showed a slight recovery after the 9th week, they remained on the subclinical/clinical levels until the end of the 12-week follow-up period. The MPAS values, on the other hand, showed a reverse trend by declining between the 6th and 9th weeks. Subjects in the two U-shaped clusters differed significantly in EPDS T1–T3, but not in terms of their anamnesis data. Compared to the stable low baseline MPAS/EPDS cluster, the U-shaped clusters had increased risk profiles (see Table 2). Cluster 1 showed increased EPDS and decreased MPAS values, and increased baby blues, but no other significantly different risk factors. There were no differences between the U-shaped clusters with respect to the risk factors.

### 3.10. Trajectories of PPD Cases in the MPAS/EPDS Clusters

All PPD cases were assigned to the two U-shaped clusters (Figure 3a,b) with the majority of PPD diagnoses (62%) being assigned to the upright U-shaped cluster, which showed subclinical EPDS at weeks 6 and 9. Its trajectory was elevated to clinical EPDS levels at weeks 3 and 12. Thus, the PPD cases belonging to this cluster showed a slight fluctuation in depressive symptoms, although the symptoms were present throughout the follow-up period. In the inverted U-shaped cluster, EPDS values increased and remained clinical after the 6th week. As regards the MPAS values, while they slightly declined over time in the upright U-shaped cluster, in the inverted U-shaped cluster they increased.

### 3.11. Trajectories of AD Cases in the MPAS/EPDS Clusters

The AD trajectories were examined in the U-shaped clusters and MPAS/EPDS cluster 1 (Figure 3c,d). Most AD diagnoses (43%) were seen in the upright U-shaped cluster with their EPDS values reaching clinical levels only at week 3 and receding thereafter. During the follow-up, an increase in MPAS values was seen in the AD cases in the upright U-shaped cluster, the same values increasing also in the AD group of the inverted U-shaped cluster, from low to subclinical/clinical levels at week 6, and returning to low levels thereafter. While MPAS remained on average around 85, the EPDS values of the MPAS/EPDS cluster 1 remained below 10 on average throughout the follow-up, and further reduced from slightly higher levels toward the 12th week. The MPAS values increased during the follow-up period.

### 3.12. Trajectories of HC in the MPAS/EPDS Clusters

Finally, as shown in Figure 3e,f, the HC trajectories were considered in all clusters with the majority of them (55%) being found in the MPAS/EPDS cluster 2, followed by cluster 1 with 40% of all HCs, clusters 1 and 2 taken as baseline clusters. The remaining 5% of HCs were shared by the two U-shaped clusters. The two baseline clusters differed in EPDS and MPAS values, with EPDS being significantly lower and MPAS higher in cluster 1. Furthermore, cluster 1 had stronger PMS and more baby blues than cluster 2. HCs belonging to the two U-shaped clusters (*n* = 22 in total; 12% of the upright U-shaped cluster and 21% of the inverted U-shaped cluster) differed from the PPD and AD cases only in EPDS values. Compared to the HCs in the baseline cluster, HCs in the upright U-shaped cluster had significantly higher EPDS values after the third week; however, they remained less than 10 on average at all time points. Other variables were not significantly different. On the other hand, HCs in the inverted U-shaped cluster had significantly higher EPDS values at all time points. Moreover, MPAS was significantly lower in the inverted U-shaped cluster, and HCs in the inverted U- shaped cluster suffered from more PMS.

## 4. Discussion

PPD has long-term effects on the wellbeing of both mother and child, potentially leading to depression becoming chronic [37]. While AD does not meet the criteria for depression at any time point, it can also lead to reduction in the quality of life [39]. An early detection of these disorders is important to mitigate the potentially high emotional and financial burden of these conditions. We investigated the dynamics of depressive mood and mother–child attachment in a large cohort of 598 women observed for a period of 12 postpartum weeks. Unlike previous research in the field, our study adopted a standardized as well as a fine-mesh follow-up approach, which, based on short-term observation of the participants’ mood changes, enabled us to detect early-onset deviations from healthy behavior. Additionally, a clinical interview was performed after 12 weeks independent of the follow-up, representing what is considered the gold standard for PPD and AD diagnostics. At week 12, based on the clinical interview, 9% of participants were diagnosed with PPD and 15% with AD. There were no significant differences between the AD and PPD groups based on risk profiles, except for income level, which was found to be lower in PPD, indicating that similar mechanisms may be involved in the development of initial depressive symptoms in both clinical groups (Appendix A). For instance, compared to HC, both AD and PPD reported experiencing baby blues more often and having a history of PMS as well as family or personal history of psychiatric disorders (Appendix A). Thus, an increased vulnerability to hormonal changes, which is characteristic of PPD and can be reflected in strongly expressed baby blues [3,19,27,45,46,47] or PMS, was also seen in our sample of postpartum AD [3,48]. However, despite similar risk profiles, the PPD and AD groups showed differences in temporal development. From the 6th postpartum week, EPDS and MPAS were significantly different between AD and PPD with EPDS being lower and MPAS higher in AD (Appendix A). The initially low attachment scores were found to increase in AD, while PPD maintained low attachment levels throughout the study. However, the differences between AD and HC in terms of MPAS and EPDS remained significant at all time points (Appendix A), suggesting that AD also deserves more clinical attention.

Based on the EPDS time courses assessed five times during the follow-up, six trajectories were identified. The baseline cluster (cluster 5), which consisted of 97% of healthy women and 72% of all study participants, had stable and low EPDS at all time points. Barring the baseline EPDS cluster, trajectories associated with depressed mood were seen to be either improving (three recovering clusters) or worsening (two deteriorating clusters). About 18% of the sample showed recovering behavior, which was reflected in clinical or subclinical levels of EPDS values only in the first few postpartum weeks, approaching baseline levels by the end of the follow-up. Seventy-nine percent of all clinically diagnosed AD cases were in the three recovering clusters (cluster 1, cluster 3, and cluster 6). The majority of participants with AD (65% of all cases) showed subclinical or clinically increased EPDS values only in the first 3 weeks of the follow-up period. In addition to AD, the improving clusters also contained a small percentage of HC (7% of all HC) and PPD (11% of all PPD) cases, the latter, similar to the AD cases in the recovering clusters, showed amelioration over time. The recovering clusters did not differ in terms of risk profiles (Appendix A). Compared to the baseline cluster, however, the recovering clusters were characterized by a higher risk profile, which was dominated by previous depression and more frequent occurrences of baby blues. In addition, cluster 1 (subclinical EPDS levels in the first 3 postpartum weeks) and cluster 6 (clinically elevated EPDS levels between weeks 3 and 6) had higher numbers of birth-related psychological and physical traumas compared to the baseline cluster. Cluster 1 and cluster 6 contained 57% of all AD cases in the sample. The increased numbers of psychological and physical traumas in this sample supports a reactive role of depressive symptoms in some AD cases.

Recovering trajectories of depressed mood have also been reported in other publications. In [22], a group of women demonstrated quickly self-remitting symptoms of depression (23% of the cohort), peaking around the 2nd postpartum week. By week 3 postpartum, for most of the women in this cluster, the symptoms had disappeared. In [20], a spontaneous resolution of depressive symptoms in 6% of the initially depressed participants was reported by 7–8 weeks. Because these studies assessed depression only by means of self-report questionnaires, PPD and AD remained clinically undifferentiated. Therefore, it is unclear if the self-remitting symptomology in these studies represented brief episodes of PPD or whether the condition in question was AD instead of PPD. In contrast to previous studies, our work demonstrates AD diagnosis to be a main characteristic of these recovering clusters. Notably, it shows that, in some cases, PPD (11% of all PPD cases in the sample) can also be self-remitting and short-lived.

The two deteriorating clusters (clusters 2 and 4) accounted for 10% of the whole cohort. Containing 84% of all PPD cases, these two clusters showed an increase in EPDS values during the 12-week period. Notably, while cluster 4 (64% PPD, 20% AD, 16% HC) demonstrated a gradual increase in EPDS values from baseline/subclinical levels to subclinical/clinical levels around the 3rd week, cluster 2 (100% PPD) had clinically elevated EPDS levels from the beginning of the follow-up, showing a further worsening of the symptoms. The PPD group belonging to cluster 2 demonstrated the highest levels of depressive mood, which, importantly, had already been elevated immediately following childbirth. In addition, cluster 2 (compared to cluster 4) was found to be associated with the highest risk profiles, such as lower income and more experience of psychiatric diagnosis in previous pregnancy. Thus, almost half of the PPD cases assigned to cluster 2 had psychiatric diagnosis in previous pregnancy compared to 3% in cluster 4 (Appendix A). The deteriorating clusters also contained a small number of AD cases, demonstrating that, in some cases, AD can also show deterioration in depression, although the symptom levels remain subclinical. Thus, clinicians should be attentive to these exceptional cases; however, they can be discovered only through an early follow-up.

Deteriorating clusters have been detected in other studies as well, i.e., in [22], the two deteriorating clusters represent 25% of the cohort, while in other samples [20,23,42], the values vary between 4%, 6.2%, and 8.2%. In [20,22], it was shown that depressive symptoms before childbirth, given their tendency to worsen after childbirth, are particularly unfavorable with respect to long-term prognosis.

Finally, based on the combination of EPDS and MPAS values, four MPAS/EPDS clusters were defined, out of which, two baseline MPAS/EPDS clusters showed very similar and stable trajectories of EPDS and MPAS scores. The EPDS scores did not reach subclinical or clinical levels at any time point in either baseline cluster. There were, however, some differences between the two baseline clusters with the first one being more homogeneous (99% HCs) and characterized by lower levels of baby blues or PMS symptoms, as well as lower EPDS and higher MPAS scores, compared to the second cluster (84% HC, 16% AD). Described as deteriorating (upright U-shaped) and recovering (inverted U-shaped) clusters, the two clinically significant clusters consisted mostly of PPD and AD patients, with a few HCs. Thus, the MPAS/EPDS clusters reflected a strong inverted relationship between MPAS and EPDS scores, higher EPDS leading to lower mother–child attachment, and vice versa.

As regards the limitations of the study, the lack of an MPAS assessment at T0 (shortly after delivery) resulted in only the MPAS/EPDS trajectories at the beginning of the 3rd postpartum week being taken into account. Furthermore, no assessment of EPDS and MPAS was performed during pregnancy. The trajectory estimation pre- and postpartum represents an interesting scientific challenge, potentially yielding a clearer differentiation of disorder trajectories based on the onset time (before, in pregnancy, or postpartum). A longer follow-up after diagnosis may lead to a better understanding of the longitudinal development of the detected trajectories.

On the whole, PPD, AD, and HCs demonstrated non-uniform trajectories. Unlike AD, PPD was found to be largely characterized by a relative deterioration of symptoms, although the trajectories of PPD cases lacked homogeneity. In particular, women with PPD, and clinically elevated EPDS scores shortly after delivery, showed the highest levels of depressed mood in the follow-up, and had the highest risk profiles for PPD, including history of PPD in previous pregnancies [49,50]. The results of previous studies and those of ours underscore the need for an early recognition of such potentially severe cases, which likely start during pregnancy. The recovering clusters, on the other hand, not only contained the majority of AD cases, they also represented higher proportions of trauma experience involving childbirth, suggesting that the temporal worsening of mood has a reactive character and is linked to childbirth-related circumstances.

As regards the EPDS clusters, the baseline cluster (cluster 5) and the deteriorating cluster (cluster 2) were the most homogeneous, consisting almost 97% of HCs and 100% of PPD cases, respectively. This indicates that the clusters become more homogeneous when their characteristics become more pronounced (in this case, consistently low or consistently high EPDS scores over the entire observational period). Thus, the trajectories with high EPDS scores correlated with higher risk profiles and higher numbers of PPD diagnosis. On the other hand, borderline cases (only subclinical or temporarily increased EPDS levels at one or two time points) constitute more inhomogeneous clusters in terms of diagnosis. For instance, even though the three recovering clusters contained the majority of AD cases (79% of all clinically diagnosed AD cases), similar patterns of improvement were seen in PPD (11% of all PPD cases), and even in HC (6% of all HC cases), although the EPDS levels were usually lower in HC. At the same time, elevated subclinical EPDS levels at all time points, with a worsening tendency (cluster 4), characterized light PPD (64% of the cluster), more prolonged AD (20% of the cluster), and, at times, even the behavior of HCs (16% of the cluster). This proves that an accurate clinical separation between light depression and AD is difficult to accomplish. Finally, we found EPDS to be highly correlated with MPAS, and two of the MPAS/EPDS clusters with elevated EPDS and reduced MPAS scores aided our classification by containing all PPD cases. Thus, our data demonstrate a dynamic and reciprocal relationship between depressed mood and maternal sensitivity.

Taken together, our results show that, frequently, PPD is not only not self-remitting, but it also tends to deteriorate, underscoring the importance of early recognition of true PPD cases and their differentiation from AD. In the clinical routine, an early follow-up at several time points should facilitate the identification of the exceptional cases.

A strong association between the trajectories and risk factors has also been observed, which may prove relevant in future research involving the possible prediction of the development of the most severely affected cluster. However, further research is needed to help establish a standardized and effective approach for early recognition based on a combination of longitudinal assessment of depressive mood and the risk factors. In addition, a combination of risk factors and diverse trajectories can be used for more precise diagnoses. In this context, another noteworthy conclusion of the paper is that there are, on the one hand, well-defined, mostly homogeneous trajectories, and mixed clusters on the other. While in the well-defined clusters, severe PPD cases are found in the deteriorating cluster, and AD cases in the recovering one, in the mixed clusters, PPD and AD remain somewhat indistinguishable from one another, which in some cases can be observed in clinical practice. Notably, more severely expressed features or symptoms lead to more precise diagnosis. A clear understanding of the inhomogeneous nature of postpartum mood deterioration is therefore critically important for the identification of more affected cases. Ours is the first study to simultaneously assess mother–child attachment and depression independent of the clinical diagnosis. Conducted as many as four times during the follow-up, this simultaneous assessment demonstrated a temporal longitudinal and reciprocal relationship between the two parameters.

## Figures and Tables

**Figure 1 jpm-12-00538-f001:**
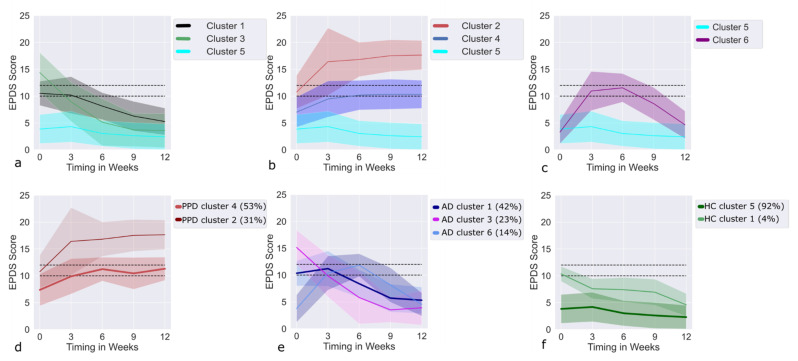
Trajectories of EPDS clusters acquired from the LCMM evaluation. Average value of the EPDS score for each cluster is shown by a solid line, shaded area represents standard deviation around the average. (**a**) Trajectories of the recovering clusters (clusters 1,3) and baseline cluster (cyan, cluster 5); (**b**) Trajectories of the deteriorating clusters (clusters 2,4) and baseline cluster (cyan, cluster 5); (**c**) Trajectories of the delayed-onset recovery cluster (clusters 6) and baseline cluster (cyan, cluster 5); (**d**–**f**) trajectories of PPD (**d**), AD (**e**) and HC (**f**) diagnoses separately. Percentages next to clusters show the percentage of the cluster in each diagnosis. Abbreviations: EPDS, Edinburgh Postnatal Depression Scale; LCMM, latent class mixed model; PPD, postpartum depression; AD, adjustment disorder; HC, healthy control.

**Figure 2 jpm-12-00538-f002:**
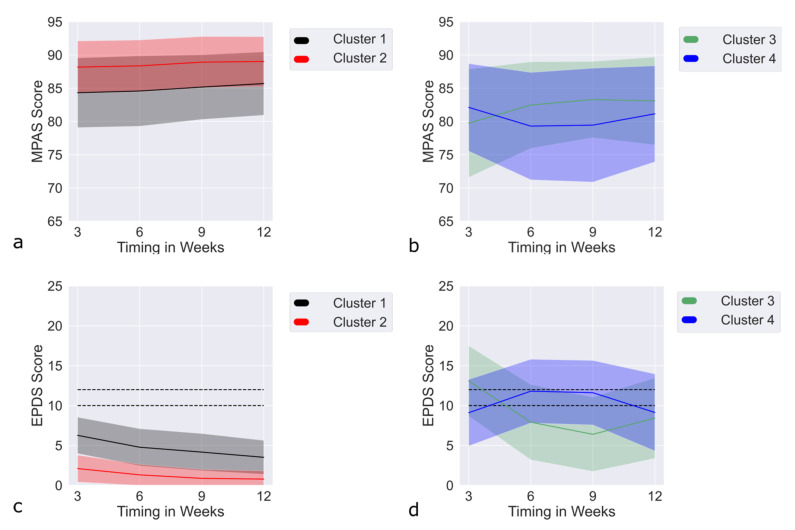
Trajectories of MPAS-EPDS clusters. (**a**,**b**) MPAS and (**c**,**d**) EPDS trajectories of multlcmm model with four clusters. Subfigures (**a**,**c**) represent trajectories of the two baseline clusters, which mostly consist of HC, while (**b**,**d**) show the U-shaped clusters, with improving and deteriorating EPDS (see text). These clusters demonstrate a strong relationship between MPAS and EPDS. Abbreviations: EPDS, Edinburgh Postnatal Depression Scale; MPAS, Maternal Postnatal Attachment Scale; HC, healthy control.

**Figure 3 jpm-12-00538-f003:**
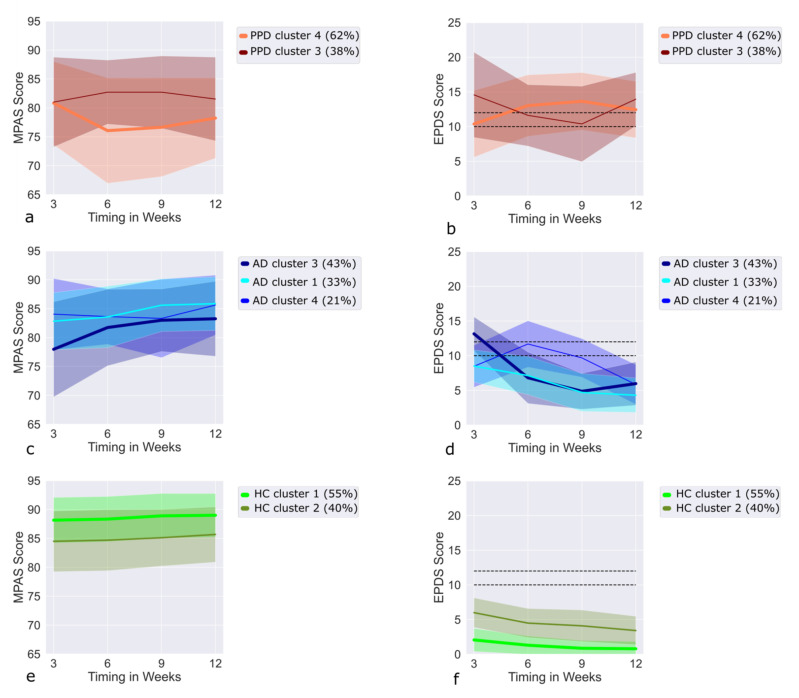
MPAS and EPDS trajectories for every diagnosis outcome, based on the latent clusters, acquired from the multlcmm clustering for: (**a**,**b**) PPD patients; (**c**,**d**) AD patients; (**e**,**f**) healthy women. Percentages next to clusters show the percentage of the cluster in each diagnosis. Abbreviations: PPD, postpartum depression; AD, postpartum adjustment disorder; HC, healthy control.

**Table 1 jpm-12-00538-t001:** Comparison of risk factors between the recovering, deteriorating, and baseline clusters.

Risk Factors	Cluster 1 vs. Baseline	Cluster 3 vs. Baseline	Cluster 6 vs. Baseline	Cluster 2 vs. Baseline	Cluster 4 vs. Baseline
Family psychiatric history	X				
Psychiatric diagnosis in previous pregnancy	X			X	X
Relocation to another ward	X				
Baby blues	X		X	X	X
Birth-related psychological and physical traumas	X		X		
Previous depression	X	X		X	X
PMS Severity			X	X	X
EPDS T0	X	X		X	X
EPDS T1	X	X	X	X	X
EPDS T2	X	X	X	X	X
EPDS T3	X		X	X	X
EPDS T4	X		X	X	X
MPAS T1	X		X	X	X
MPAS T2	X			X	X
MPAS T3	X			X	X
MPAS T4		X		X	
SLE				X	X
Support at home				X	X
Family status				X	
Income				X	

Abbreviations: PMS, premenstrual syndrome; EPDS, Edinburgh Postnatal Depression Scale; MPAS, Maternal Postnatal Attachment Scale; SLE, stressful life events. X means, that significant difference was discovered.

**Table 2 jpm-12-00538-t002:** Risk factor comparison of the clusters with the baseline MPAS/EPDS cluster.

Risk Factors	Cluster 1 MPAS/EPDS vs. Baseline	Upright U-shaped vs. Baseline	Inverted U-shaped vs. Baseline
Baby blues	X	X	X
Previous depression		X	X
PMS Severity	X	X	X
EPDS T0	X	X	X
EPDS T1	X	X	X
EPDS T2	X	X	X
EPDS T3	X	X	X
EPDS T4	X	X	X
MPAS T1	X	X	X
MPAS T2	X	X	X
MPAS T3	X	X	X
MPAS T4	X	X	X
SLE			X
Support at home		X	X
Birth-related psychological and physical traumas		X	X
Psychiatric diagnosis in previous pregnancy			X
Family psychiatric history		X	

Abbreviations: MPAS, Maternal Postnatal Attachment Scale; EPDS, Edinburgh Postnatal Depression Scale; PMS, premenstrual syndrome; SLE, stressful life events. X means, that significant difference was discovered.

## Data Availability

The data presented in this study are available on request from the corresponding author. The data are not publicly available due to privacy restrictions.

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
