# Peer review of "Characterization of Depressive Symptom Trajectories in Women between Childbirth and Diagnosis"

_jpm, 2022, doi:10.3390/jpm12040538_

Round 1

Reviewer 1 Report

[JPM] Manuscript ID: jpm-1604417 

Title: Characterization of Depressive Symptom Trajectories in 2 Women between Childbirth and Diagnosis

Comments

This paper describes a large cohort of 598 young mothers and measured their depressed condition and mother-child attachment after childbirth, at 3 weeks’ interval, and until 12 weeks postpartum. Clinical interviews were conducted to obtain the diagnoses of postpartum depression, postpartum adjustment disorder, or healthy controls. From their depressive symptom trajectories,the majority of women with postpartum depression showed deteriorating, and the majority of adjustment disorder cases improved. Higher EPDS levels correlated with the higher risk factor profiles. The four MPAS/EPDS clusters reflected that higher EPDS lead to lower mother-child attachment.  Overall, the manuscript is technically sound and the statistical analysis was performed appropriately. I would like to provide some comments that needed authors’ revisions to further explain the clinical implications of their results: 

  1. Although the terms and epidemiological descriptions of postpartum blues, postpartum adjustment disorder, and postpartum depression have been defined in the introduction section, please provide an explanation what they planned to do was different from past literature regarding clinical application or implication. For example, on Line 92, authors explained that past studies vary in time points, and authors aimed to focus on the short-term period of early postpartum period. Why is focusing on this particularly short term period important? 
  2. Similarly, Line 96, why is focusing on postpartum adjustment disorder as a separate diagnosis important clinically? 
  3. Line 97, the authors mentioned that information about risk factors were frequently missing. Why is focusing on these information important clinically? If these are important, why didn’t past studies include these risk factors? Did the authors find results supporting these importance at the end of the day? I believe explanations are still needed. 
  4. In the Discussion section, the authors should also compare and explain the the possible mechanisms of how their results different from the past research regarding design, concepts, measures, time frame, and main results. 
  5. Most importantly, what are the clinical implications of their results? If the difference between postpartum depression and postpartum adjustment disorder was that one showed deterioration and one showed improvement, well, this seems to be how the diagnosis was made by criteria. Their result only demonstrates further confirmation that in those that deteriorated, there were higher percentages of PPD instead of AD. Was there a causal relationship? or How would knowing this fact help the clinical situation from the perspective of early recognition or prevention? 
  6. Similar clinical gap was also shown when considering that higher EPDS levels correlated with the higher risk factor profiles. We may already know from past research that women with PPD had higher EPDS levels than women with AD, and for the sake of early recognition, regular measure of EPDS might already be helpful. Then please explain what other clinical implications do the results from this paper further add ?
  7. Regarding the finding that ‘The four MPAS/EPDS clusters reflected that higher EPDS lead to lower mother-child attachment.’ Please explain in the Discussion section, what were the comparisons with past literature? I believe there were already literature that showed inverse relationship between MPAS and EPDS. Then what other clinical implications do the results from this paper further add ?Please look through relevant literature to see if their current results have been reported (summarize in the Introduction section), or may be supported by past study results (summarize in the Discussion section).
  8. Could these results or the application of their results really help early recognition of PPD?
  9. There might still need a short summary regarding why is it important to identify PPD earlier, if this is one of the purpose to conduct this study. If this is important, why is it important as well to differentiate AD? 
  10. On Line 538, they said this proved the clinical separation between depression and AD is difficult to accomplish. So what does the result from this paper add to real-world clinical practice?
  11. To make this research more influential to its potential readers of the JPM, please explain further how significant PPD is affecting the health of both mother and child (Line 58). 
  12. Possible mechanisms between depression (measured by EPDS and clinical interview), and mother-child attachment, and distributions of different clusters may need more elaboration or comparisons with previous studies. The current Discussion section mainly re-summarize their results, without further discussions on comparisons, possible mechanisms, clinical or research implications. Since this is the main point of the study, I’m somewhat confused why there weren’t any contents regarding more in- depth discussion than just summarizing the results again. 
  13. The paper is unbelievably lengthy, is there any way to shorten it for the reader?

Author Response

This paper describes a large cohort of 598 young mothers and measured their depressed condition and mother-child attachment after childbirth, at 3 weeks’ interval, and until 12 weeks postpartum. Clinical interviews were conducted to obtain the diagnoses of postpartum depression, postpartum adjustment disorder, or healthy controls. From their depressive symptom trajectories, the majority of women with postpartum depression showed deteriorating, and the majority of adjustment disorder cases improved. Higher EPDS levels correlated with the higher risk factor profiles. The four MPAS/EPDS clusters reflected that higher EPDS lead to lower mother-child attachment.  Overall, the manuscript is technically sound and the statistical analysis was performed appropriately. I would like to provide some comments that needed authors’ revisions to further explain the clinical implications of their results.

Answer: We thank the reviewer for these remarks. The manuscript has been revised in a way that brings, among other things, the clinical implications of our results into sharper focus.  Following are our replies to the reviewer’s comments. 

  1. Question: Although the terms and epidemiological descriptions of postpartum blues, postpartum adjustment disorder, and postpartum depression have been defined in the introduction section, please provide an explanation what they planned to do was different from past literature regarding clinical application or implication. For example, on Line 92, authors explained that past studies vary in time points, and authors aimed to focus on the short-term period of early postpartum period. Why is focusing on this particularly short term period important?

Answer: It is a pertinent question. In an effort to explain the importance of close observation in the early postpartum period, we have revised the relevant part of the introduction (see page 3, lines 104-112) in the following manner:

“The early postpartum period (10 to 19 days after childbirth in particular) is a time of the highest risk for women to develop mental disorders [1], with the most frequent one being PPD. Women with PPD are at an increased risk of developing chronic psychiatric conditions. For instance, 50% of women with a history of PPD have been found to develop at least one further episode of depression [2]. Therefore, it is critically important to monitor women’s health within the first few weeks of childbirth, beginning as close to the point of delivery as possible. Early detection, intervention, and appropriate treatment of PPD can help prevent chronification, reducing the emotional and financial burden of the disease [3].”

  1. Question: Similarly, Line 96, why is focusing on postpartum adjustment disorder as a separate diagnosis important clinically?

Answer: We have sought to explain the clinical importance of the diagnosis of adjustment disorder (AD) in greater detail on page 3, lines 115-126:

“AD is recognized as a stress-response syndrome. As a subthreshold diagnosis, it is defined as a maladaptive reaction to an identifiable stressor (DSM-V). Thus, the crucial difference between postpartum AD and PPD is that the severity of the AD symptoms does not meet the criteria for depression at any time point. Therefore, in clinical practice, both baby blues and AD need to be considered as important differential diagnoses of PPD. Neither condition has the tremendously debilitating effect of a clinically manifest depression and, normally, neither requires treatment. A timely differentiation between PPD and AD can aid the identification of the most affected and vulnerable cases. However, subclinical depression or adjustment disorder may, in some cases, be a significant public health issue because of its high prevalence in the population [4]. In some cases, women suffering from these conditions are likely to benefit from relevant supportive treatment.”

  1. Question: Line 97, the authors mentioned that information about risk factors were frequently missing. Why is focusing on these information important clinically? If these are important, why didn’t past studies include these risk factors? Did the authors find results supporting these importance at the end of the day? I believe explanations are still needed.

Answer: We thank the reviewer for directing our attention to this particular point. An explanation regarding the importance of the risk factors has now been added to the introduction on page 3, lines 129-131. In the results and the discussion, we have sought to demonstrate a strong association between the discovered clusters and risk factors (Tables 1 and 2, page 8, lines 337-346, page 9, lines 354-362, page 13, lines 485-489, 511-516, page 14, lines 539-545), suggesting that the risk factors would be predictive of the most severe cases in particular. In addition, the risk is higher when the number of factors is larger. The lack of any information vis-a-vis risk factors is, therefore, a limitation, because it is on the basis of the risk factors that women affected by depression can be better identified. This helps explain the limitation of some of the previous studies. To what degree risk factors can predict who are most likely to be affected by depression requires to be adequately investigated to facilitate an early recognition of PPD. This is an additional focus of our research related to early recognition of PPD.

The following piece of text has been added:

„Risk factors provide an insight into the possible development of PPD due to their association with the prevalence of the diagnoses. The presence of multiple risk factors is a likely indication of stronger depressive symptoms. “

  1. Question: In the Discussion section, the authors should also compare and explain the possible mechanisms of how their results different from the past research regarding design, concepts, measures, time frame, and main results.

Answer: We have revised the discussion to clarify the differences between our paper and the available literature (see page 3, lines 135-137 and page 13, lines 475-480).

Page 3: ”The follow-up was conducted by means of, on the one hand, a fine-mesh approach and, on the other, an independent clinical interview performed after 12 weeks. This two-pronged approach facilitated unbiased diagnoses of PPD and AD.”

Page 13: ”Unlike previous research in the field, our study adopted a standardized as well as a fine-mesh follow-up approach, which, based on short-term observation of the participants’ mood changes, enabled us to detect early-onset deviations from healthy behavior. Additionally, a clinical interview was performed after 12 weeks independent of the follow-up, representing what is considered the gold standard for PPD and AD diagnostics.”

  1. Question: Most importantly, what are the clinical implications of their results? If the difference between postpartum depression and postpartum adjustment disorder was that one showed deterioration and one showed improvement, well, this seems to be how the diagnosis was made by criteria. Their result only demonstrates further confirmation that in those that deteriorated, there were higher percentages of PPD instead of AD. Was there a causal relationship? or How would knowing this fact help the clinical situation from the perspective of early recognition or prevention?

Answer: While we did indeed find AD to be mostly self-remitting, and PPD showing a deteriorating tendency, we also observed some exceptions. Some PPD cases, diagnosed by the clinician independently of the trajectory, were found to be self-remitting, while some AD cases were found to be deteriorating. These exceptions were also seen in non-depressed women. The diagnoses of AD and PPD were based on the clinical interview, the major characteristic feature of AD being that its symptom severity does not meet the criteria for PPD. Later, based on the EPDS trajectories, we saw that those cases were also more frequently self-remitting. In this context, it needs to be borne in mind that this self-remitting characteristic was not used as a diagnostic criterion for AD. Given that PPD is frequently not self-remitting and has a deteriorating tendency, an early recognition of true PPD cases and their differentiation from AD is of crucial importance. In clinical practice, an early follow-up at several time points should aid the identification of exceptional cases. We have added this piece of information to the concluding part of discussion (page 16, lines 603-624).

Also, clinicians should be mindful of any inhomogeneous development of the disorders (page 15, lines 545-546):

“Thus, clinicians should be attentive to these exceptional cases, which can be identified only through an early follow-up.”

  1. Question: Similar clinical gap was also shown when considering that higher EPDS levels correlated with the higher risk factor profiles. We may already know from past research that women with PPD had higher EPDS levels than women with AD, and for the sake of early recognition, regular measure of EPDS might already be helpful. Then please explain what other clinical implications do the results from this paper further add ?

Answer: As mentioned above, a longitudinal follow-up is necessary for proper diagnostics. In addition, a combination of risk factors and diverse trajectories can be used for more precise diagnoses. We have added this piece of information to the concluding part of discussion (page 16, lines 603-624).

  1. Question: Regarding the finding that ‘The four MPAS/EPDS clusters reflected that higher EPDS lead to lower mother-child attachment.’ Please explain in the Discussion section, what were the comparisons with past literature? I believe there were already literature that showed inverse relationship between MPAS and EPDS. Then what other clinical implications do the results from this paper further add ?Please look through relevant literature to see if their current results have been reported (summarize in the Introduction section), or may be supported by past study results (summarize in the Discussion section).

Answer: To the best of our knowledge, our study is the first to simultaneously assess the longitudinal MPAS-EPDS relationship in the early postpartum period. In Hahn et al., 2021, we described this relationship in detail. There is hardly any other study that has done this. While [5] investigates the relationship between mother-child attachment and women’s mood, they acquired data only in two time points after childbirth. We, on the other hand, concentrated on the postpartum period and a clinical diagnosis in relationship to the simultaneous behavior of MPAS and EPDS. Other studies either focused on pre-birth bonding [6], or conducted their investigations after the onset of depression [7], [8].

Please, refer to the following piece of text in the discussion, page (p.16, lines 603-625).

  1. Question: Could these results or the application of their results really help early recognition of PPD?

Answer: We observed a strong association between EPDS values up to the sixth week and the trajectories until the end of the follow-up. A robust link was also seen between the trajectories and the risk factors. In future research, this observation may prove useful in predicting the development of the most severely affected cluster before it is firmly established by the 3rd month. This potential reduction in detection time would be enormously helpful to the affected women. That being said, further research is needed to establish a standardized and effective approach for early recognition based on a combination of longitudinal assessment of depressive mood and the risk factors. We have added this piece of information to the concluding part of discussion (page 16, lines 603-624).

  1. Question: There might still need a short summary regarding why is it important to identify PPD earlier, if this is one of the purpose to conduct this study. If this is important, why is it important as well to differentiate AD?

Answer: We thank the reviewer for this suggestion, based on which we have added the following piece of text to page 13, lines 469-473:

“PPD has long-term effects on the wellbeing of both mother and child, potentially leading to depression becoming chronic [2]. While AD does not meet the criteria for depression at any time point, it can also lead to the reduction in the quality of life [4]. An early detection of these disorders is important to mitigate the potentially high emotional and financial burden of these conditions.”

  1. Question: On Line 538, they said this proved the clinical separation between depression and ADis difficult to accomplish. So what does the result from this paper add to real-world clinical practice?

Answer: We are afraid our point is slightly misconstrued here. We do not say that it is difficult to clinically separate PPD from AD. All we say is that some borderline cases, such as mild depression, cannot be easily differentiated from AD, which is why we stress the necessity of a longitudinal observation. This problem of separating mild forms of PPD from AD would be familiar to most clinical practitioners.

One of the main results of the paper is that there are, on the one hand, well-defined, mostly homogeneous trajectories, and mixed clusters on the other. While in the well-defined clusters, severe PPD cases are seen in the deteriorating cluster, and AD cases in the recovering one, in the mixed clusters PPD and AD behave in a similar fashion. A clear understanding of the inhomogeneity of the diagnoses would facilitate the treatment of each disorder.

  1. Question: To make this research more influential to its potential readers of the JPM, please explain further how significant PPD is affecting the health of both mother and child (Line 58).

Answer: We have added the following piece of text to the introduction on page 2, lines 60-68:

PPD influences a wide range of outcomes for offspring, including infant growth, physical health, nutritional status, executive functions, socioemotional development, academic achievement, as well as the risk for psychopathology [9]–[16]. In particular, there is an increased risk for a transmission of intergenerational depression [16]–[19]. Therefore, early bio-markers, predictors and early interventions are important for the wellbeing of both mothers and their children. However, the reality is that PPD is often overlooked during the postnatal visits, resulting  in lost time and opportunities as preventative interventions are most effective if administered shortly after delivery [20]–[22].”

  1. Question: Possible mechanisms between depression (measured by EPDS and clinical interview), and mother-child attachment, and distributions of different clusters may need more elaboration or comparisons with previous studies. The current Discussion section mainly re-summarize their results, without further discussions on comparisons, possible mechanisms, clinical or research implications. Since this is the main point of the study, I’m somewhat confused why there weren’t any contents regarding more in- depth discussion than just summarizing the results again.

Answer:

The point is well taken. In the revised manuscript, the discussion is concluded (p.16, lines 603-625) by the most salient results of the study being summarized in the following manner:

“Taken together, our results show that, frequently, PPD is not only not self-remitting, but it also tends to deteriorate, underscoring the importance of early recognition of true PPD cases and their differentiation from AD. In the clinical routine, an early follow-up at several time points should facilitate the identification of the exceptional cases.

A strong association between the trajectories and risk factors has also been observed, which may prove relevant in future research involving the possible prediction of the development of the most severely affected cluster. However, further research is needed to help establish a standardized and effective approach for early recognition based on a combination of longitudinal assessment of depressive mood and the risk factors. Thus, a longitudinal follow-up is important for proper diagnostics. In addition, a combination of risk factors and diverse trajectories can be used for more precise diagnoses. In this context, another noteworthy conclusion of the paper is that there are, on the one hand, well-defined, mostly homogeneous trajectories, and mixed clusters on the other. While in the well-defined clusters, severe PPD cases are found in the deteriorating cluster, and AD cases in the recovering one, in the mixed clusters, PPD and AD remain somewhat indistinguishable from one another, which in some cases can be observed in clinical routine. Notably, more severely expressed features or symptoms lead to more precise diagnosis. A clear understanding of the inhomogeneous nature of postpartum mood deterioration is therefore critically important for the identification of more affected cases. Ours is the first study to simultaneously assess mother-child attachment and depression independent of the clinical diagnosis. Conducted as many as four times during the follow-up, this simultaneous assessment demonstrated a temporal longitudinal and reciprocal relationship between the two parameters.”

  1. Question: The paper is unbelievably lengthy, is there any way to shorten it for the reader?

Answer: We agree with the reviewer, and have tried to shorten the length of the manuscript, but, given the extensive background information that is needed to demonstrate the importance and clinical relevance of the research, we have not, we must admit, quite succeeded in our efforts.

[1]        T. Munk-Olsen, T. M. Laursen, C. B. Pedersen, O. Mors, and P. B. Mortensen, “New Parents and Mental Disorders: A Population-Based Register Study,” JAMA, vol. 296, no. 21, p. 2582, Dec. 2006, doi: 10.1001/jama.296.21.2582.

[2]        C. L. Taylor, M. Broadbent, M. Khondoker, R. J. Stewart, and L. M. Howard, “Predictors of severe relapse in pregnant women with psychotic or bipolar disorders,” J. Psychiatr. Res., vol. 104, pp. 100–107, Sep. 2018, doi: 10.1016/j.jpsychires.2018.06.019.

[3]        M. Aron Halfin, “Depression: The Benefits of Early and Appropriate Treatment,” Suppl. Featur. Publ., vol. 13, no. 4 Suppl, Nov. 2007, Accessed: Mar. 04, 2022. [Online]. Available: https://www.ajmc.com/view/nov07-2638ps092-s097

[4]        H. Akil et al., “Treatment resistant depression: A multi-scale, systems biology approach,” Neurosci. Biobehav. Rev., vol. 84, pp. 272–288, Jan. 2018, doi: 10.1016/j.neubiorev.2017.08.019.

[5]        H. Ohoka et al., “Effects of maternal depressive symptomatology during pregnancy and the postpartum period on infant-mother attachment: Depression and infant-mother attachment,” Psychiatry Clin. Neurosci., vol. 68, no. 8, pp. 631–639, Aug. 2014, doi: 10.1111/pcn.12171.

[6]        J. T. Condon, “The Spectrum of Fetal Abuse in Pregnant Women:,” J. Nerv. Ment. Dis., vol. 174, no. 9, pp. 509–516, Sep. 1986, doi: 10.1097/00005053-198609000-00001.

[7]        P. M. Dietz, S. B. Williams, W. M. Callaghan, D. J. Bachman, E. P. Whitlock, and M. C. Hornbrook, “Clinically Identified Maternal Depression Before, During, and After Pregnancies Ending in Live Births,” Am. J. Psychiatry, vol. 164, no. 10, pp. 1515–1520, Oct. 2007, doi: 10.1176/appi.ajp.2007.06111893.

[8]        I. Brockington, “Postpartum psychiatric disorders,” The Lancet, vol. 363, no. 9405, pp. 303–310, Jan. 2004, doi: 10.1016/S0140-6736(03)15390-1.

[9]        E. D. Barker, N. Kirkham, J. Ng, and S. K. G. Jensen, “Prenatal maternal depression symptoms and nutrition, and child cognitive function,” Br. J. Psychiatry, vol. 203, no. 6, pp. 417–421, Dec. 2013, doi: 10.1192/bjp.bp.113.129486.

[10]      A. Rahman, P. J. Surkan, C. E. Cayetano, P. Rwagatare, and K. E. Dickson, “Grand Challenges: Integrating Maternal Mental Health into Maternal and Child Health Programmes,” PLoS Med., vol. 10, no. 5, p. e1001442, May 2013, doi: 10.1371/journal.pmed.1001442.

[11]      C. Monk, M. K. Georgieff, and E. A. Osterholm, “Research Review: Maternal prenatal distress and poor nutrition - mutually influencing risk factors affecting infant neurocognitive development: Maternal prenatal distress and poor nutrition,” J. Child Psychol. Psychiatry, vol. 54, no. 2, pp. 115–130, Feb. 2013, doi: 10.1111/jcpp.12000.

[12]      R. M. Pearson et al., “Maternal perinatal mental health and offspring academic achievement at age 16: the mediating role of childhood executive function,” J. Child Psychol. Psychiatry, vol. 57, no. 4, pp. 491–501, Apr. 2016, doi: 10.1111/jcpp.12483.

[13]      S. H. Goodman, M. H. Rouse, A. M. Connell, M. R. Broth, C. M. Hall, and D. Heyward, “Maternal Depression and Child Psychopathology: A Meta-Analytic Review,” Clin. Child Fam. Psychol. Rev., vol. 14, no. 1, pp. 1–27, Mar. 2011, doi: 10.1007/s10567-010-0080-1.

[14]      L. Murray, A. Arteche, P. Fearon, S. Halligan, T. Croudace, and P. Cooper, “The effects of maternal postnatal depression and child sex on academic performance at age 16 years: a developmental approach: PND & child cognitive and academic outcomes,” J. Child Psychol. Psychiatry, vol. 51, no. 10, pp. 1150–1159, Oct. 2010, doi: 10.1111/j.1469-7610.2010.02259.x.

[15]      H. Shen et al., “Associations of Parental Depression With Child School Performance at Age 16 Years in Sweden,” JAMA Psychiatry, vol. 73, no. 3, p. 239, Mar. 2016, doi: 10.1001/jamapsychiatry.2015.2917.

[16]      M. M. Weissman et al., “Treatment of Maternal Depression in a Medication Clinical Trial and Its Effect on Children,” Am. J. Psychiatry, vol. 172, no. 5, pp. 450–459, May 2015, doi: 10.1176/appi.ajp.2014.13121679.

[17]      M. M. Weissman, P. Wickramaratne, Y. Nomura, V. Warner, D. Pilowsky, and H. Verdeli, “Offspring of Depressed Parents: 20 Years Later,” Am. J. Psychiatry, vol. 163, no. 6, pp. 1001–1008, Jun. 2006, doi: 10.1176/ajp.2006.163.6.1001.

[18]      R. M. Pearson et al., “Maternal Depression During Pregnancy and the Postnatal Period: Risks and Possible Mechanisms for Offspring Depression at Age 18 Years,” JAMA Psychiatry, vol. 70, no. 12, p. 1312, Dec. 2013, doi: 10.1001/jamapsychiatry.2013.2163.

[19]      J. Yang, P. Yin, D. Wei, K. Wang, Y. Li, and J. Qiu, “Effects of parental emotional warmth on the relationship between regional gray matter volume and depression-related personality traits,” Soc. Neurosci., vol. 12, no. 3, pp. 337–348, May 2017, doi: 10.1080/17470919.2016.1174150.

[20]      L. J. Miller, “Postpartum Depression,” JAMA, vol. 287, no. 6, p. 762, Feb. 2002, doi: 10.1001/jama.287.6.762.

[21]      M. W. O’Hara, “Postpartum depression: what we know,” J. Clin. Psychol., vol. 65, no. 12, pp. 1258–1269, Dec. 2009, doi: 10.1002/jclp.20644.

[22]      M. W. O’Hara and J. E. McCabe, “Postpartum Depression: Current Status and Future Directions,” Annu. Rev. Clin. Psychol., vol. 9, no. 1, pp. 379–407, Mar. 2013, doi: 10.1146/annurev-clinpsy-050212-185612.

Reviewer 2 Report

In the manuscript titled „Characterization of Depressive Symptom Trajectories in Women Between Childbirth and Diagnosis", the authors present the useful information for this issue. It is important to reduce and prevent postpartal depression in women. 

Delete the word "in" in the sentence "Qandil and colleagues in [9] focused on the heterogeneity ..." 

Minor checking of grammar is suggested.

Author Response

In the manuscript titled „Characterization of Depressive Symptom Trajectories in Women Between Childbirth and Diagnosis", the authors present the useful information for this issue. It is important to reduce and prevent postpartal depression in women.

Delete the word "in" in the sentence "Qandil and colleagues in [9] focused on the heterogeneity ..."

Minor checking of grammar is suggested.

 We thank the reviewer for this generous comment. The correction has been duly made and the manuscript properly edited.

Round 2

Reviewer 1 Report

Many thanks! My concerns have all be very-well addressed.